# Cerebral Blood Deoxygenation by a Postural Change Detected by Near-Infrared Spectroscopy Has a Close Association with Cerebral Infarction

**DOI:** 10.3390/brainsci12101419

**Published:** 2022-10-21

**Authors:** Hiroshi Irisawa, Naoki Inui, Takashi Mizushima, Hiroshi Watanabe

**Affiliations:** 1Department of Clinical Pharmacology and Therapeutics, Hamamatsu University School of Medicine, Hamamatsu 431-3192, Japan; 2Department of Rehabilitation Medicine, Dokkyo Medical University, Shimotsugagun 321-0293, Japan; 3Department of Rehabilitation Medicine, Enshu Hospital, Hamamatsu 430-0929, Japan

**Keywords:** cerebrovascular circulation, spectroscopy, near-infrared, cerebral infarction, risk factor

## Abstract

Background: The recent introduction of near-infrared spectroscopy has enabled the monitoring of cerebral blood flow in real-time. Previous studies have shown that blood flow velocity is a predictor of cardiovascular disease. We hypothesized that cerebral oxygenation with a change in posture is a predictor for cerebral infarction. We designed a cross-sectional study to investigate the relationship between postural-related changes in cerebral oxygenation and a history of chronic cerebral infarction. Methods: A total of 100 consecutive participants were enrolled in this study. We evaluated changes in cerebral oxygenation with a change in posture from the supine to the upright position in the bilateral forehead. The association between a decline in cerebral oxygenation and chronic cerebral infarction was analyzed with multiple logistic regression adjusted for covariates. Results: Cerebral blood oxygenation increased in 52 participants and decreased in 48 participants with a postural change. The prevalence of decreased cerebral oxygenation was 76.3% in participants with chronic cerebral infarction. Multiple logistic regression analysis showed that a decline in cerebral oxygenation upon a postural change was strongly associated with chronic cerebral infarction (adjusted odds ratio: 3.42, *p* = 0.025). Conclusions: Cerebral blood oxygenation upon a postural change could be a useful predictor for cerebral infarction.

## 1. Introduction

Cerebral infarction is one of the most common causes of death and is the main cause of persistent and acquired disability in adults worldwide. Considering demographic changes, a further increase in cerebral infarction rates is expected. Moreover, cerebral infarction is expected to increasingly affect younger patients. The World Health Organization refers to cerebral infarction as the incoming epidemic of the 21st century. Therefore, knowing the risk factors for stroke is of prime importance. On the other hand, measurements of cerebral blood flow and oxygenation provide important information, which predicts neurological outcomes in patients with acute cerebral infarction and schizophrenia and those undergoing cardiac surgery [1,2,3]. Cerebral blood flow can be monitored by several methods, including single-photon emission computed tomography (SPECT), positron emission tomography, functional magnetic resonance imaging, ultrasonic Doppler blood flow, and near-infrared spectroscopy (NIRS). NIRS noninvasively measures hemoglobin (Hb) levels in the brain using near-infrared rays. Compared with other technologies, NIRS has several advantages as follows: (1) it allows flexible measurements in sitting, standing, and moving subjects, (2) it is a radiation-based, completely noninvasive technique that does not cause adverse effects on the body in repeated measurements, even in children, (3) it has a high resolution, and (4) it is compact and movable. The use of NIRS, such as cerebral oxygen monitors, is increasing in the medical field. This increase in use is because such systems are simple but enable observation of changes in brain activity over time via monitoring of Hb levels, which reflect fluctuations in regional cerebral blood flow. Furthermore, the NIRS system is relatively less expensive as a brain function imaging device, and its running cost is low. Well-known applications of NIRS include monitoring cerebral blood flow and hypoxic conditions in several clinical settings [2,4,5].

Some investigators have shown that bipolar disorder patients show unique changes in cerebral blood flow when they perform tasks [1,6]. Furthermore, other investigators have shown that a change in posture also affects blood flow and cerebral oxygenation [7,8,9]. These findings suggest that emotional and physical stress leads to a change in cerebral blood flow in the fetus and healthy individuals. Previous pilot studies have shown that cerebral oxygenation is changed in relation to positioning [10], and systemic desaturations are rapidly followed by local cerebral desaturation in patients with acute cerebral infarction [11,12]. If this hypothesis is correct, measuring cerebral blood flow during postural changes may help determine the risk of future stroke. In addition, Pase et al. suggested that blood flow velocity could be a predictor of cardiovascular disease [13].

Therefore, we hypothesized that cerebral oxygenation with a change in posture is a predictor for cerebral infarction. If this hypothesis is correct, measuring cerebral blood flow during postural changes may help determine the risk of future stroke. We hypothesized that cerebral blood flow should decrease during postural changes in patients with stroke and designed a cross-sectional study to investigate the relationship between postural-related changes in cerebral oxygenation and a history of chronic cerebral infarction.

## 2. Materials and Methods

### 2.1. Ethics Statement

This study protocol complied with the Declaration of Helsinki and was approved by the institutional research review board of Hamamatsu University School of Medicine, Hamamatsu, Japan (Approval ID: 22-5-1). Written informed consent was provided by all participants. The study was registered at the UMIN Clinical Trials Registry (UMIN000004669).

### 2.2. Study Participants

The study initially included 126 consecutive patients who received post-stroke or post-fracture rehabilitation treatment at Enshu Hospital (Hamamatsu, Japan) between April 2013 and March 2014. Patients unable to remain standing and those who had severe cognitive impairment, severe dysphasia and orthostatic hypotension were excluded from the study, and finally, 100 were evaluated.

### 2.3. Procedure

Cerebral oxygenation was continuously evaluated by a NIRS device (TRS-10, Hamamatsu Photonics K.K., Hamamatsu, Japan). The TRS-10 system produces 3 light bundles with wavelengths of 759, 797, and 833 nm [14]. We measured oxy-Hb, deoxy-Hb, and t-Hb in the bilateral forehead. Because the TRS-10 system has a single channel, 2 consecutive measurements were conducted in each participant for measurement of cerebral oxygenation of the right side of the forehead followed by the left side. An optode, which has infrared light irradiation and reception probes in a single device, was fixed on the participant’s head with Velcro and a headband so that the irradiated infrared light was positioned at Fp1 and Fp2 according to the International 10–20 system. This NIRS device can measure the oxygenation of tissues within a semicircular area between the irradiation and reception probes. The measurement depth increases as the distance between the irradiation and reception probes increases, but the limit is 5 cm deep because the farther the distance, the weaker the light reaching the reception probe. In a study reporting simultaneous measurements with NIRS and positron emission tomography, NIRS measurements with irradiation and reception probes 4 cm apart were correlated with positron emission tomography measurements that were 0.9 cm deep from the brain surface [15]. Based on this finding, we set the distance between the irradiation and reception probes to 4 cm outside of the infrared reception port on the optode.

### 2.4. Cerebral Oxygenation Measurement Protocol

Because a strong correlation between oxy-Hb and regional cerebral blood flow was previously reported [16,17], oxy-Hb concentrations were sampled every second in this study.

Participants were lying down in a dark room. Baseline measurement of oxy-Hb was conducted for 1 min after resting in the supine position for 4 min. The participant was then asked to stand in an upright position. We used the same wording and timing at the change in posture in all participants because language stimulation is known to alter cerebral blood flow [18]. All of the participants stood up by themselves without assistance. Post-standing oxy-Hb was measured for 1 min after standing for 2 min. Systemic blood pressure was also measured in the supine and standing positions using an automatic hemodynamometer (HBP1300; Omron Corp., Tokyo, Japan).

The mean values of changes in oxy-Hb (Δoxy-Hb) of the right and left sides of the forehead were calculated. A cut-off point of mean Δoxy-Hb < 0 was applied as an index of a decline in cerebral oxygenation.

### 2.5. Measurements

Each participant was present at the laboratory on the morning of the sampling day after fasting for 8–12 h. Each participant rested in a seated position prior to sampling. Blood sampling was performed by experienced laboratory technicians. Venous blood was collected into 5 mL serum separator tubes (Venoject II; Terumo, Tokyo, Japan) and 2 mL EDTA tubes (Venoject II; Terumo, Tokyo, Japan). EDTA samples immediately and serum tubes after 30 min in an upright position. All collection tubes were centrifuged at 3000× *g* for 10 min. Blood cell counts and hemoglobin A1c from EDTA samples were measured with an automated analyzer (Celltac G; Nihon Kohden, Tokyo, Japan). Levels of triglycerides, high-density lipoprotein cholesterol, total cholesterol, and uric acid were measured from serum using an automated analyzer (LABOSPECT006; Hitachi High-Tech Co., Tokyo, Japan). Systolic blood pressure and diastolic blood pressure were measured with the participants in the supine position. Height and body weight were measured with the participants wearing lightweight clothing. Global cognitive function was assessed by the Mini-Mental State Examination. The mean intima-media thickness of the common carotid artery was measured as an index of arteriosclerosis by sonography (Logiq 5; GE Healthcare, Little Chalfont, UK).

### 2.6. Statistical Analysis

Data are expressed as the mean ± standard deviation (SD). We divided the study participants into 2 groups according to fluctuation patterns in cerebral oxygenation. Categorical variables were compared between the groups of participants by the chi-square test. Differences between groups were examined by unpaired *t*-test. Multiple linear regression analysis was used to assess the independent variables that affected a decline in cerebral oxygenation. The association of a decline in cerebral oxygenation and chronic cerebral infarction was analyzed with a multiple logistic regression model adjusted for covariates (Table 1). A *p*-value < 0.05 was regarded as statistically significant. The computations were performed using the IBM SPSS Statistics program (version 21.0; IBM Corporation, New York, NY, USA).

## 3. Results

The age of the study participants ranged from 20–96 years, with a mean of 71.1 ± 4.0 years. Thirty-eight participants (17 men and 19 women) had a history of cerebral infarction (mean age, 72.8 ± 11.0 years; duration from onset to the time of measurement of cerebral oxygenation, 17.0 ± 13.7 weeks). Table 2 shows the underlying disease of the study participants.

### 3.1. Postural Change and Fluctuations in Cerebral Oxygenation

Upon a change in posture, 52 participants showed increased cerebral oxygenation, and 48 participants showed decreased cerebral oxygenation with the upright position. The prevalence of decreased cerebral oxygenation was higher in participants with chronic cerebral infarction than in those without chronic cerebral infarction (76.3% (*n* = 29) vs. 30.6% (*n* = 19), *p* < *0*.001). Typical patterns of fluctuating cerebral oxygenation are shown in Figure 1. No participants demonstrated a lack of fluctuation of cerebral oxygenation and had opposite results (i.e., both an increase and a decrease for the right and left measurements). None of the participants with or without a history of chronic cerebral infarction showed a significant difference in the fluctuation of cerebral blood flow between the affected and unaffected sides. After 2 min in the standing position, the patient was returned to the supine position, and measurements continued; after 10 min, all patients returned to baseline values.

### 3.2. Factors Affecting a Decline in Cerebral Oxygenation 

Table 3 shows the baseline characteristics of the study participants. The proportions of chronic cerebral infarction and male sex in the decreased cerebral oxygenation group were higher than those in the increased cerebral oxygenation group, but these differences were not significant (*p* = 0.077). The white blood cell count was significantly lower in the increased cerebral oxygenation group than in the decreased cerebral oxygenation group (*p* < 0.05). No other baseline demographic and clinical characteristics differed between the 2 groups. The partial regression coefficients of participants’ characteristics that affected a decrease in cerebral oxygenation are shown in Table 4. Chronic cerebral infarction was found to be an independent variable that decreased cerebral oxygenation.

### 3.3. A Decline in Cerebral Oxygenation Is Independently Associated with Chronic Cerebral Infarction

Because chronic cerebral infarction was selected as an independent variable for the decrease in cerebral oxygenation, we finally investigated the association between a decline in cerebral oxygenation and chronic cerebral infarction. Multiple logistic regression analysis showed that a decline in cerebral oxygenation in response to the upright posture was strongly associated with chronic cerebral infarction (Table 5). The text continues here.

## 4. Discussion

The present study evaluated fluctuations in cerebral oxygenation with a change in posture from the supine to the upright posture using a NIRS device and assessed its predictive value for the prevalence of chronic cerebral infarction. We found the following: (1) there was no significant difference in the fluctuation of cerebral blood oxygenation between the affected and unaffected sides in participants with chronic cerebral infarction (Table 3); (2) multiple linear regression analysis showed that a history of chronic cerebral infarction was an independent variable that decreased cerebral oxygenation (Table 4); and (3) a decline in cerebral oxygenation in response to an upright posture has a close association with a history of chronic cerebral infarction (Table 5).

There has been speculation that cerebral circulation might fluctuate as the body position changes. However, no reports have verified this phenomenon until recently because of practical issues with the measurement method of cerebral blood flow. Studies on human cerebral hemodynamics started to become more common after 1990 when transcranial Doppler sonography was developed by Aaslid et al. [19]. In a previous transcranial Doppler sonography study, the middle cerebral artery mean flow velocity increased by 14% in the head-down tilt position [20]. Satake et al. used SPECT to measure regional cerebral blood flow in the head-down tilt position, and they also observed significantly increased blood flow at the basal ganglia and the cerebellum [21]. Although these reports indicated that cerebral blood flow could fluctuate by the head-down tilting position, the body position examined in these studies is not commonly used in daily life. Fluctuations in cerebral blood flow with a standing upright posture, which is part of daily life, have not been previously evaluated because transcranial Doppler sonography and SPECT were not suitable for cerebral blood flow measurements in the standing position. In contrast, a NIRS device can continuously measure cerebral oxygenation, which strongly correlates with regional cerebral blood flow in moving participants. This feature of NIRS enabled cerebral oxygenation monitoring upon changes in posture in our study. A previous study indicated that a high proportion of patients with a history of cerebral infarction had a decline in cerebral blood flow with a change in posture, and even a certain proportion of healthy subjects demonstrated decreased cerebral blood flow [7]. It is also possible that cerebral vasospasm after acute subarachnoid hemorrhage is associated with decreased cerebral blood flow, as seen in this study [22,23]. Furthermore, orthostatic changes in cerebral oxygenation detected by NIRS are reproducible [24], and this reproducibility was confirmed by our preliminary study (data not shown).

This study has some limitations. (1) The potential contribution of skin blood flow to Δoxy-Hb has been identified as an issue [25], and this possibility cannot be ruled out in our determination of oxy-Hb using the modified Beer–Lambert law. A previous study demonstrated a correlation between Δoxy-Hb and fluctuations in cerebral blood flow [14]. Another study also showed small changes in oxy-Hb levels by clamping the external carotid artery to the scalp and cranium during carotid endarterectomy [26], suggesting a limited effect on skin blood flow. (2) Some of our participants were taking vasodilator agents, which may have affected the results. Although we did not evaluate the effect of oral medications that were taken by the participants at the time of the study, our preliminary study indicated that oral antihypertensive agents were unlikely to affect fluctuations in cerebral blood flow (data not shown). (3) We did not have any information on lifestyle factors, including smoking status and alcohol-drinking behavior. The lack of these data might have partially affected our results. (4) We cannot explain the mechanisms for a decrease in cerebral oxygenation with a change in posture from our results. However, a decline in cerebral oxygenation upon a change in posture is not likely due to cerebral arterial occlusion or stenosis because this phenomenon is observed on both the stroke-affected and unaffected sides of the brain. (5) Because we used a single-channel NIRS system, we could only investigate cerebral blood flow within a narrow range of the prefrontal cortex. Previous studies have indicated that an increase in cerebral blood flow in one region causes a decrease in other regions [27,28,29]. Multi-channel NIRS is currently mainstream and should be used in future studies to understand fluctuations in cerebral blood flow in the entire brain upon postural changes. (6) Changes in brain tissue over time after brain injury, such as hydrocephalus, have received increasing attention in recent years [30,31]. In this study, it is also possible that the response of cerebral blood flow may differ depending on the duration of time after cerebral infarction. (7) The number of subjects in our study is small, and we need to consider increasing the number of subjects in the future.

## 5. Conclusions

In conclusion, a decline in cerebral oxygenation in response to an upright posture has a close association with a history of chronic cerebral infarction. This phenomenon could be useful for risk stratification of cerebral infarction. A large prospective study on this issue is required in the future.

## Figures and Tables

**Figure 1 brainsci-12-01419-f001:**
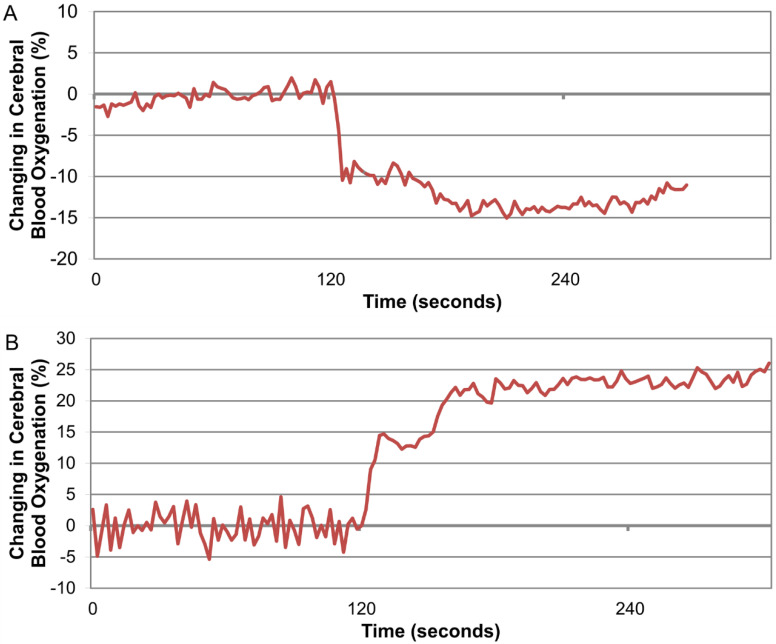
Fluctuations in cerebral blood flow in the supine position (0–120 s) followed by the standing position. (**A**) Cerebral blood flow was increased upon standing. (**B**) Cerebral blood flow was decreased upon standing.

**Table 1 brainsci-12-01419-t001:** Definitions of covariates for their effect on fluctuations in cerebral oxygenation.

Data Entry	Definitions of Covariates
Advanced age	Age > 75 years old
Obese	Body mass index > 25 kg/m^2^
Hypertriglyceridemia	Triglyceride > 150 mg/dL
Hypercholesterolemia	Total cholesterol > 220 mg/dL
Hypo HDL cholesterolemia	High-density lipoprotein cholesterol < 40 mg/dL
Hyperuricemia	Uric acid > 7.0 mg/dL
Hypertension	Blood pressure > 140/90 mmHg
Diabetes	HbA1c > 6.1%
Carotid artery thickening	Intima-media thickness > 1.1 mm
Anemia	Hemoglobin < 13.5 g/L in men or <11.5 g/L in women

Abbreviations: HDL cholesterol, high-density lipoprotein cholesterol; HbA1c, hemoglobin A1c.

**Table 2 brainsci-12-01419-t002:** The underlying disease of the study participants.

Underlying Disease	*n*
Atheromatous cerebral infarction	24
Lacunar infarction	12
Cerebral embolism	2
Lower extremity fractures	56
Post abdominal surgery	6

**Table 3 brainsci-12-01419-t003:** Baseline clinical characteristics of the study participants.

	Increased Cerebral Oxygenation	Decreased Cerebral Oxygenation	*p* Value
Participants, *n*	52	48	
History of chronic cerebral infarction, *n*	9 (17.3)	29 (60.4)	<0.001
Age (year)	70.0 ± 15.0	72.3 ± 12.8	0.411
Male gender	19 (36.5)	26 (54.2)	0.077
Percent change in cerebral oxygenation, %	4.1 ± 2.2	−5.9 ± 5.1	<0.001
Participants with cerebral infarction: percent change in cerebral oxygenation on the unaffected side, %	2.8 ± 2.6	−6.2 ± 3.8	<0.001
Participants with cerebral infarction: percent change in cerebral oxygenation on the affected side, %	3.1 ± 2.9	−6.8 ± 4.2	<0.001
Mean intima-media thickness (mm)	0.86 ± 0.21	1.03 ± 0.36	0.006
Systolic blood pressure (spine) (mmHg)	118.1 ± 14.2	117.7 ± 13.2	0.899
Diastolic blood pressure (spine) (mmHg)	64.5 ± 10.0	66.0 ± 10.8	0.473
Systolic blood pressure (upright) (mmHg)	116.1 ± 19.3	112.1 ± 18.6	0.364
Diastolic blood pressure (upright) (mmHg)	65.8 ± 14.2	69.6 ± 12.3	0.146
Body height (cm)	155.1 ± 9.4	157.4 ± 8.5	0.191
Body weight (kg)	51.5 ± 9.7	53.4 ± 10.6	0.346
BMI (kg/m^2^)	21.3 ± 2.9	21.4 ± 3.1	0.871
Hb (g/dL)	11.9 ± 1.7	12.4 ± 1.8	0.171
White blood cell count (10^3^/μL)	5.2 ± 1.2	5.7 ± 1.4	0.048
Platelet count (10^4^/μL)	22.0 ± 6.8	21.6 ± 5.7	0.739
Triglycerides (mg/dL)	112.2 ± 37.7	120.8 ± 58.9	0.385
Total cholesterol (mg/dL)	169.0 ± 25.9	172.6 ± 43.9	0.626
HDL cholesterol (mg/dL)	43.5 ± 11.4	46.7 ± 11.2	0.158
Uric acid (mg/dL)	5.1 ± 1.3	5.5 ± 1.5	0.103
HbA1c (%)	5.5 ± 0.7	5.7± 0.7	0.261
MMSE	25.9 ± 4.7	24.1 ± 5.7	0.098

Data are expressed as *n* (%) or mean ± standard deviation. Abbreviations: HDL cholesterol, high-density lipoprotein cholesterol; HbA1c, hemoglobin A1c; Hb, hemoglobin; MMSE, Mini-Mental State Examination; BMI, body mass index.

**Table 4 brainsci-12-01419-t004:** Partial regression coefficients of variables that affected fluctuations in cerebral oxygenation.

Variables	Unstandardized Coefficients (B)	Standardized Coefficients (β)	95% CI for B	*p* Value
Chronic cerebral infarction	−0.049	−0.374	−0.078, −0.019	0.002
Age	1.74 × 10^−5^	0.004	−0.001, 0.001	0.978
Male gender	−0.001	−0.010	−0.032, 0.030	0.935
Systolic BP	−3.86 × 10^−5^	−0.008	−0.001, 0.001	0.936
Hemoglobin	−0.002	−0.051	−0.010, 0.008	0.707
Triglyceride	1.55 × 10^−4^	−0.120	4.58 × 10^−4^, 1.49 × 10^−4^	0.314
Total cholesterol	1.07 × 10^−4^	0.060	2.68 × 10^−4^, 4.83 × 10^−4^	0.571
HDL cholesterol	−0.001	−0.123	−0.002, 4.84 × 10^−4^	0.247
Uric acid	−0.003	−0.078	−0.013, 0.006	0.483
HbA1c	0.001	0.010	−0.018, 0.020	0.927
Mean IMT	−0.028	−0.134	−0.081, 0.025	0.296
MMSE	0.001	0.073	−0.002, 0.003	0.500

Abbreviations: BP, blood pressure; HbA1c, hemoglobin A1c; IMT, media-intima thickness: MMES, Mini-Mental State Examination; HDL cholesterol, high-density lipoprotein cholesterol.

**Table 5 brainsci-12-01419-t005:** Adjusted associations between chronic cerebral infarction and decreased cerebral oxygenation.

Variables	Adjusted Odds Ratios	95% CI	*p* Value
Decreased cerebral oxygenation	3.42	(1.17, 10.02)	0.025
Advanced age	0.41	(0.13, 1.28)	0.124
Male gender	1.10	(0.39, 3.10)	0.860
Obese	0.96	(0.19, 4.93)	0.963
Hypertriglyceridemia	2.28	(0.34, 15.47)	0.398
Hypercholesterolemia	0.68	(0.16, 2.93)	0.603
Hypo HDL cholesterolemia	0.28	(0.98, 0.82)	0.020
Hyperuricemia	0.76	(0.13, 4.31)	0.752
Hypertension	4.21	(0.49, 36.02)	0.189
Diabetes	0.65	(0.12, 3.39)	0.606
Carotid artery thickening	4.42	(1.25, 15.58)	0.021
Anemia	0.36	(0.13, 1.03)	0.056

Abbreviations: CI, confidence interval; HDL cholesterol, high-density lipoprotein cholesterol.

## Data Availability

The data that support the findings of this study are available from the corresponding author, [H.I.], upon reasonable request.

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
