# Peer review of "Cerebral Blood Deoxygenation by a Postural Change Detected by Near-Infrared Spectroscopy Has a Close Association with Cerebral Infarction"

_brainsci, 2022, doi:10.3390/brainsci12101419_

Round 1

Reviewer 1 Report

The authors present an interesting study examining the relationship between postural-related changes in cerebral oxygenation and history of infarction. Briefly, the authors recruited 100 patients at random and subjected them to examination with an NIRS device to determine cerebral oxygenation. Examination of these indices in the patients in a standing and supine position was measured, in addition to bloods analyses. Together, the authors suggest postural change coincides with decrease in cerebral oxygenation and increased association with cerebral infarction.

In reviewing the manuscript I had a number of concerns. The following should be addressed by the authors when preparing a suitable revision.

1.       While the introduction does a good job of introducing the idea of measuring cerebral blood flow in the context of cerebral infarction, there is scope to improve the background on the link between posture, cerebral blood flow, and infarction. I felt this was a bit light the writing could be improved in justifying the grounds for this study.

2.       The type of rehabilitation the patients/participants of the study are partaking in is not mentioned, nor is it clear what their backgrounds are. Clarity over how they were recruited/selected would be appreciated.

3.       It would have been idea to recruit a control group to compare the indices measured. Why was this not considered?

4.       What system/assays were used to profile the bloods?

5.       Did the authors examine blood oxygenation over time? While they profile changes in the first 2 minutes of positional change, did they examine if the change returned to baseline or persisted?  

Author Response

Thank you for your careful peer review. Our team discussed and revised the paper according to the reviewers' comments in order to make it a better paper.

  1. Thank you for pointing this out. We added a sentence to line 59 of the Introduction to better emphasize the purpose.
  2. Thank you for pointing this out. We have detailed the participant selection section.
  3. The purpose of this study was to examine factors that influence postural changes in cerebral blood flow, and the results showed that having a history of cerebral infarction reduced cerebral blood flow. The control group would be, in other words, the group without cerebral infarction.
  4. The blood tests used in this study were of a general nature, blood counts and biochemistry, and we have not described the methods we used.
  5. Thank you for pointing this out. We have added a sentence on line 162.

Reviewer 2 Report

Interesting paper looking at cerebral blood deoxygenation associated with postural change. 

The BMI and metabolic syndrome is interesting and should be expanded.

The physiologic data is interesting and relevant. Should further expand on concept of potential vasospasm PMID: 36081602 and DCI PMID: 35537284. 

If the above concepts are addressed and suggested references added, could be of interest. 

Author Response

Thank you for your careful peer review. Our team discussed and revised the paper according to the reviewers' comments in order to make it a better paper.

We followed the reviewer's advice and mentioned the association between cerebral blood flow and vasospasm based on the references in the discussion section. (Line 237)

Reviewer 3 Report

The authors used 100 patients to evaluate changes in cerebral oxygenation with a change in posture from the supine to the upright position in the bilateral forehead. They found an association between a decline in cerebral oxygenation and cerebral infarction with a logistic regression adjusted for covariates. Multiple logistic regression analysis showed that a decline in cerebral oxygenation upon a postural change was strongly associated with chronic cerebral infarction. The novelty of this paper is not clear and this is one of the big problems of this paper. In addition, the discussion section needs considerable correction. I prefer to have the response of the authors on the below comments.

1.       “We hypothesized that cerebral oxygenation with a change … cerebral infarction”. Do you have a doubt about it and want to hypothesize …? this is obviously proved previously in some papers such as [https://doi.org/10.1016/j.mvr.2010.02.004]. The novelty of this paper is not clear.

2.       “The prevalence of decreased cerebral oxygenation was 76.3% in participants with 22 chronic cerebral infarction” I think the sample size of this study is not enough to report prevalence as one of your outputs. You can report it in your paper but you mention this as one of your main findings in the abstract section!

3.       In the first paragraph of the Introduction, please add some sentences about the population of “cerebral infarction” based on previous papers and the related statistics about it to a better understanding of readers about the importance and necessity of this study.

4.       The last paragraph of the Introduction should be totally corrected. The hypothesis and its novelty are not clear. Please write the clear gap of previous studies and your novelty.

5.       “2.2. Study participants”: you must report all exclusion and inclusion criteria in detail in this section.

6.       “2.3. Procedure”: the resolution and accuracy of the equipment and tools should be added.

7.       “Statistical analysis”. You had some parameters that depend on each other. Should you use the corrected Bonferroni p-value? why?

8.       The result section is well-designed but some of the results were missed in the discussion section. When you didn't discuss many of your results in the Discussion section and you gave up them, this means from the beginning reporting these results was not valuable.

9.       Data validation and comparing your results with previous results are not enough and you should improve this section of the discussion.

10.   You should add some ideas for future studies at the end of the last paragraph of the Discussion section. Recent studies showed the primary facts about the effect of changes in biomechanical parameters on cerebral infarction [10.3389/fbioe.2022.900644] and [10.1371/journal.pone.0196216]. You can discuss it and suggest it for future studies. 

Author Response

Thank you for your careful peer review. Our team discussed and revised the paper according to the reviewers' comments in order to make it a better paper.

  1. The paper presented by the reviewer examines cerebral infarction and changes in cerebral blood flow. Our study is very different because it examines changes in cerebral blood flow due to postural changes. We have changed the text to emphasize postural changes as pointed out by the reviewer.
  2. As the reviewer pointed out, the number of subjects in our study is small. We have included that point in the paper as a limitation of our research.
  3. Thank you for pointing this out. We added a sentence to line 59 of the Introduction to better emphasize the purpose.
  4. Thank you for pointing this out. We added a sentence to line 61 of the Introduction.
  5. Thank you for pointing this out. We have detailed the participant selection section.
  6. We have added a reference describing the principles of the TRS10, the instrument we used in this study.
  7. We did not use the Bonferroni test because it is said to be less accurate when testing more than five groups.
  8. We are discussing the results obtained in the Discussion section, but have added a Discussion section to make it easier to understand which results we are discussing.
  9. We have redescribed it in the Discussion section, emphasizing cerebral blood flow variations due to postural changes.
  10. We have added a sentence that discusses the paper as presented by the reviewers.

Round 2

Reviewer 1 Report

The authors have addressed some of my comments and the manuscript is improved. I do believe however there is scope for further improvement - I dont not believe the introduction, despite the addition of the new final paragraph, suitably gives enough background to reason why this study was pursued/where it fits in the context of the current health problem. I also was disappointed that no details at all were provided on how the blood analysis was performed. As such, I am recommending this for major corrections in that I believe there are still important aspects of this article that require attention.

Author Response

Thank you for your kind remarks.

We are again convinced by the reviewer‘s points.

At the beginning of the introduction, we stated that cerebral infarction is an important topic in current health issues and that its prediction is important, which is an important background for this study.

In the Methods section, we have made major revisions to the detailed measurement methods for blood tests, as pointed out by the reviewer.

We greatly appreciate the reviewer`s helpful advice and comments on how to improve this paper.

Reviewer 2 Report

accept

Author Response

Thank you for your peer review. We thank the reviewer again for your cooperation and advice.

Reviewer 3 Report

Accept

Author Response

(The authors gave the same response as above.)
